# Analysis of Long-Term Moon-Based Observation Characteristics for Arctic and Antarctic

**Yue Sui [1,2,3], Huadong Guo [1,2], Guang Liu [1,2,\*] and Yuanzhen Ren [4]**

[1]   Aerospace information research institute, Chinese Academy of Sciences, Beijing 100094, China;
      suiyue@radi.ac.cn (Y.S.); hdguo@radi.ac.cn (H.G.)
[2]   Institute of Remote Sensing and Digital Earth, Chinese Academy of Sciences, Beijing 100094, China
[3]   University of Chinese Academy of Sciences, Beijing 100049, China
[4]   Beijing Institute of Radio Measurement, The Second Academy of China Aerospace Science and Industry
      Corporation (CASIC), Beijing 100854, China; renyuanzhen13@mails.ucas.ac.cn
[\*]   Correspondence: liuguang@radi.ac.cn; Tel.: +86-10-8217-8103

**Abstract:** The Antarctic and Arctic have always been critical areas of earth science research and are sensitive to global climate change. Global climate change exhibits diversity characteristics on both temporal and spatial scales. Since the Moon-based earth observation platform could provide large-scale, multi-angle, and long-term measurements complementary to the satellite-based Earth observation data, it is necessary to study the observation characteristics of this new platform. With deepening understanding of Moon-based observations, we have seen its good observation ability in the middle and low latitudes of the Earth's surface, but for polar regions, we need to further study the observation characteristics of this platform. Based on the above objectives, we used the Moon-based Earth observation geometric model to quantify the geometric relationship between the Sun, Moon, and Earth. Assuming the sensor is at the center of the nearside of the Moon, the coverage characteristics of the earth feature points are counted. The observation intervals, access frequency, and the angle information of each point during 100 years were obtained, and the variation rule was analyzed. The research showed that the lunar platform could carry out ideal observations for the polar regions. For the North and South poles, a continuous observation duration of 14.5 days could be obtained, and as the latitude decreased, the duration time was reduced to less than one day at the latitude of 65° in each hemisphere. The dominant observation time of the North Pole is concentrated from mid-March to mid-September, and for the South Pole, it is the rest of the year, and as the latitude decreases, it extends outward from both sides. The annual coverage time and frequency will change with the relationship between the Moon and the Earth. This study also proves that the Moon-based observation has multi-angle observation advantages for the Arctic and the Antarctic areas, which can help better understand large-scale geoscientific phenomena. The above findings indicate that the Moon-based observation can be applied as a new type of remote sensing technology to the observation field of the Earth's polar regions.

**Keywords:** Moon-based Earth observation; Antarctic and Arctic regions; geometric simulation; coverage characteristics

## 1. Introduction

Since the 1960s, various types of Earth observation satellites, space stations, deep space detectors, and other artificial equipment have been successfully launched into space internationally, and a series of satellite Earth observation programs have been carried out [1]. Aviation and aerospace Earth observation applications have formed strong technical support in the fields of climate change, resources

and environmental change detection, and disaster risk reduction and prevention [2,3]. Humans can use satellite observation technology to obtain high-precision, high-temporal-resolution data of the atmosphere, ocean, and land. With the capabilities of global scale remote sensing data coverage, it provides direct observation data and information for geoscience research [4]. These data have contributed for the better understanding of the Earth system. However, the above Earth observation methods are all based on artificial satellite platforms, and the results of remote sensing observations depend heavily on the performance of the platform [1]. However, in the course of the on-orbit operation of existing satellite platforms, although the orbit stability and sensor differences are less and less restrictive to image quality, the spatial and temporal characteristics of the imaging coverage are still limited by the orbital height and position [5]. Thus, our understanding of how some anthropogenic influences affect climate change is still incomplete [6]. Existing satellite platforms are still unable to meet the needs of co-observation of polar phenomena and global change, which has led to space constraints in the study of large-scale scientific phenomena [7]. With further development in earth system science research, the demand for understanding of the Earth has also risen to a new level, and it is necessary to develop new Earth observation techniques, systems, and platforms to meet the rising demand [8,9].

The Moon is the only natural and long-term stable satellite of the Earth. With the increasing interest in Moon exploration, many countries worldwide have implemented a lunar exploration program [10] and have placed sensors for the purpose of different applications in lunar orbit or on the lunar surface [11–17]. In the last century, the United States and the Soviet Union have carried out many lunar explorations in order to compete in space technology. These programs mainly completed the lunar soft landing and surface imaging, sample collections, and surface operations [12–15,18,19]. In particular, the far ultraviolet camera deployed on the surface of the Moon carried by Apollo 16 was the first experiment in observing the Earth from the Moon [14]. At the beginning of this century, the rest of the world began to carry out a lunar exploration program. Europe, Japan, and India all launched their lunar exploration plans in order to carry out various scientific studies on the Moon [17,20,21]. The Chinese Lunar Exploration Program (CLEP) Chang'e was officially started on March 1, 2003 [22] with the aim to launch landing detectors, establish a Moon-based observation station, and take samples back to Earth [23,24]. The exploration of lunar exploration activities laid the foundation for the development of Moon-based Earth observation.

Based on the above-mentioned lunar exploration programs that have already been launched and are being developed, we consider that the establishment of a Moon-based Earth observation station has become an inevitable trend in world space activities [9]. The idea of Moon-based Earth observation is to explore ways to set up sensors on the Moon to observe environmental changes on a global scale. Compared with artificial satellites, the Moon-based observation of the geoscience phenomenon has the advantages of long-term, stability, and consistency. This will better meet the needs of global change research, and complement the space borne Earth observations [25]. Therefore, the Moon, the only extraterrestrial body humans have ever visited, is a reasonable option for establishing an Earth observation platform [18].

Until now, various studies and experiments on Moon-based observations have been carried out. At the beginning of the 21st century, some scientific conferences on lunar exploration and development were held internationally. In February 2007, in the Workshop on Science Associated with the Lunar Exploration Architecture, some ideas of a lunar Earth observatory were proposed and several feasible scientific goals were discussed [26]. Then, Chinese scientist Guo first systematically proposed a Moon-based Earth observation project for global change study in 2009 [27]. The Moon-based Earth observation project will probe into ways of setting up sensors on the Moon's surface to make Earth observations on a global scale to study changes in the Earth system. Since then, many scientists have begun to conduct research in the field of Moon-based Earth observations. Palle discussed the possibility of a lunar-based experiment of Earth radiation budget monitoring, and evaluated its scientific advantages compared to other observing platforms [28]; Moccia [29] and Fornaro [30] proposed the

idea of Moon-based synthetic aperture radar (SAR) and preliminarily designed the parameters of the SAR system, and then discussed the possible applications. F. He focused on the radiation properties of the Earth's plasmasphere and provided a basis for the design of the Moon-based extreme ultraviolet (EUV) imager [31]. Since 2010, Guo et al. have carried out a systematic research of Moon-based observations. The research comprises early conceptual studies of Moon-based observations for global change monitoring and developed a simplified observation model that quantitatively analyzed the spatial resolution and coverage area of the Moon-based platform [32]. Zhang presented simulations of interferometers in the Earth orbit and on the lunar surface to guide the design and optimization of space-based ultra-long wavelength missions, and the results showed that the lunar regolith will have an undesirable effect on the observations [33]. Ding and Guo discussed the performance of the Moon-based SAR system and the peculiarity of interferometry, and discussed decorrelation deriving from the geometry variety [25,34]. On the 566th Workshop of Xiangshan Scientific Conference in June 2016, Guo and other scientists systematically expounded the connotation, characteristics, and potential detection fields of Moon-based Earth observation [35]. In July of the same year, the IEEE International Geoscience and Remote Sensing Symposium (IGARSS) was held in Beijing, where some experts focused on the advantages and prospects of Moon-based Earth observations from the perspective of lunar synthetic aperture radar, atmospheric observation and energy budget research, geometric research, and global scale scientific phenomena. In recent years, some scholars in Guo's team have devoted themselves to the study of the observation geometry and coverage characteristics of Moon-based platforms, and others are working on the detailed analysis of the geometric model. Ren and Guo developed the reference systems' transformation and simulation system based on the Jet Propulsion Laboratory (JPL) ephemerides and Earth orientation parameters, and defined an effective coverage parameter for assessing the ability of optical coverage [36]. Afterward, they summarized the global angular characteristics of the Moon-based Earth observations [37]. During 2016–2019, Ye and Guo established an improved geometric model and analyzed the coverage performance and observation duration of the Moon-based platform. Furthermore, they investigated the variation regularity of the looking vector direction and the relationship with the sensor's position, and later on proposed an ideal location for the Moon-based platform [10,38–42]. The result shows that Moon-based observations have large spatial coverage advantages in the study of three-polar regions and latitude between 30°N–80°N or 30°S–80°S are the ideal locations for the Moon-based Earth observation platform [10,40]. There are also some advances in the simulation study of Moon-based Earth observation applications. Xu and Chen developed a Moon-based SAR theory where an accurate curved trajectory signal model was derived [43,44]. Nie presented a method for simulating the land surface temperature measured by Moon-based Earth observations, which showed good accuracy compared to existing satellites and methods [45].

Although the former studies of our group have focused on Moon-based Earth observation geometry and the preliminary calculation of the observational characteristics of the Moon-based observation, detailed observational characteristics of the long-time polar regions under the Moon-based perspective are still limited [46]. In this article, based on the preliminary works of Guo's team, we analyzed the Moon-based sensor coverage characteristics of different locations in the Arctic and Antarctic, and provided a theoretical basis for the future study of polar scientific phenomena. The rest of this article is organized as follows. Section 2 outlines the use of the established coordinate transformation system to unify the position and attitude of the Sun, the Earth, and the Moon to the same coordinate system. Section 3 calculates the coverage characteristics and angular characteristics for different points in polar regions for a period of over 100 years. With the results, we compared the coverage characteristics of different latitudes in the North and South poles, and analyzed the causes of the differences. In Section 4, we propose a polar scientific phenomenon applicable to lunar observations based on the special characteristics obtained above.

## 2. Data and Simulation System

In order to understand the observational characteristics of Moon-based Earth observations, we need to first understand the precise geometric relationship between the Sun, the Earth and the Moon. The basis of Moon-based observation geometry research is to accurately understand the operational law, and to obtain the position and attitude information of these three celestial bodies, through which we can get the observation characteristics and angular characteristics of the Moon-based observation platform [36,37]. We know that all the celestial bodies have their own fixed orbit. By using the simulation system, we can simulate the motions of celestial bodies in space, which helps us to conduct further qualitative research. The main process is shown in Figure 1.

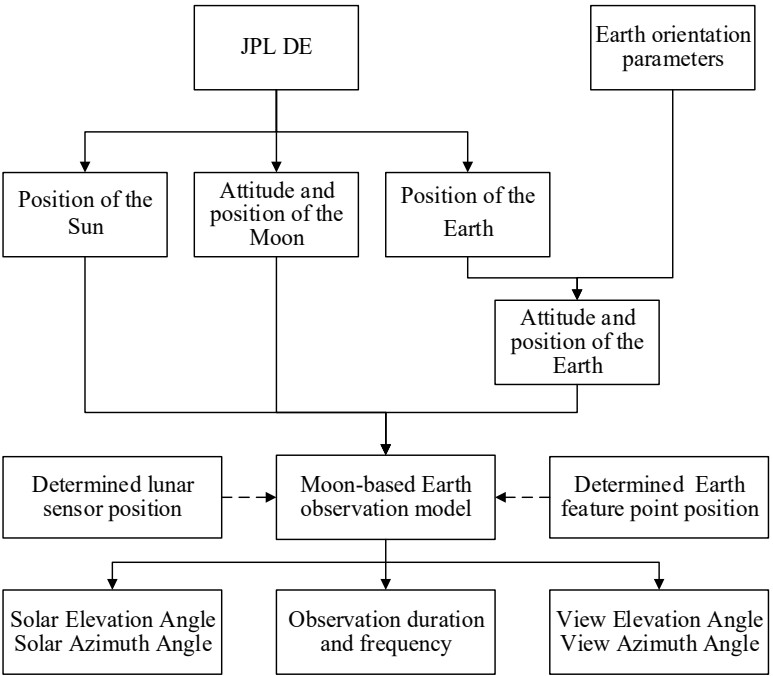

**Figure 1.** The main work content and steps of this paper.

### 2.1. JPL Ephemerides and Earth Orientation Parameters

The JPL Development Ephemeris (DE) designates one of a series of models of the solar system produced at the Jet Propulsion Laboratory in California, primarily for the purposes of spacecraft navigation and astronomy [47]. The models consist of computer representations of positions, velocities, and accelerations of major solar system bodies, covering a specified span of years [48]. The positions and velocities of the Sun, Earth, Moon, and planets, along with the orientation of the Moon, result from a numerically integrated dynamical model. Since the 1960s, there have been many versions of the JPL DE such as DE200, DE403, DE421, DE430, DE432, and so on. Among these, the DE430, which was created in 2013 covering the period from January 1, 1550 to January 22, 2650 and includes the nutation and libration, is very suitable for the study of the operating law of the Sun–Earth–Moon in this paper. The ephemeris data are available online at http://ssd.jpl.nasa.gov/.

Precise transformations between the international celestial and terrestrial reference frames are needed for many advanced geodetic and astronomical tasks including positioning and navigation on Earth and in space [49]. The Earth orientation parameter (EOP) determines the direction from the observer on Earth to the celestial body in the inertial space. Due to the influences of the Sun, the Moon, and other celestial bodies, the Earth's rotation axis pointing, in addition to the long-term slow-moving (precession), is also superimposed on a variety of smaller amplitude vibration cycles, which is called nutation (its cycle is 18.6 years). In order to accurately express the attitude of the Earth at a certain moment, the International Astronomical Union (IAU) recommends using

EOP to describe the irregular rotation of the Earth [50]. The EOP are available online at https://www.iers.org/IERS/EN/Home/home_node.html.

*2.2. Transformation between the Reference Systems*

In the study of the Moon-based platform, the Earth–Moon relative motion model must be established before the analysis of the coverage characteristics. Based on the ephemeris data and Earth orientation parameters, the reference system transformation and a simulation system of Moon-based Earth observations were developed. The transformation process includes many reference systems [36]. In addition to reference systems related to the celestial bodies, there are some other reference systems for Moon-based sensors, but they were not analyzed in this paper. In this paper, the reference system we used can be briefly divided into three categories. The first type is the geodetic reference system. The geodetic reference system is established in the geodetic survey with the reference ellipsoid as the reference plane [36]. The position of the ground point is represented by the longitude, latitude, and height. The geodetic reference system is usually a right-handed system. Typical representatives in this paper are the geographic reference system (GRS) and selenographic reference system (SRS) [36]. The second type is a reference system dealing with the parameters related to the celestial body surface. The origin is defined as the center of the celestial body. The x-axis intersects the sphere at 0° latitude and 0° longitude, the fundamental plane is at the celestial equatorial plane and the reference system rotates with the sphere such as the Moon-Centered Moon-Fixed coordinate system (MCMF) and the international terrestrial reference system (ITRS) [36,39]. The third type is the celestial reference system. The XY reference plane is a plane through the celestial sphere's center perpendicular to the axis, with the X-axis pointing to the vernal equinox. It is obvious that the celestial reference system is kind of an equatorial reference system. When the centers of celestial spheres are selected as the center of the Sun, Earth, or the Moon, three different celestial reference systems are defined, namely, the heliocentric celestial reference system (HCRS), geocentric celestial reference system (GCRS), and selenocentric celestial reference system (SCRS), respectively [36].

2.2.1. Transformation from SRS to SCRS

In order to convert the SRS to the SCRS, we needed to separate it into three steps. First, we needed to carry out the transformation from SRS to mean Earth/polar axis (ME) system. Then, we needed to transform the ME system to the principal axis (PA) system. Finally, rotation from the PA system to the SCRS needed to be performed. Among these, both the ME system and the PA system belong to the category of the Moon-Centered Moon-Fixed coordinate system (MCMF), and there was a slight difference between them. The transformation procedure is as follows:

$$
\begin{bmatrix} X_{SCRS} \\ Y_{SCRS} \\ Z_{SCRS} \end{bmatrix} = [L] \begin{bmatrix} x_{PA} \\ y_{PA} \\ z_{PA} \end{bmatrix} = [L][C] \begin{bmatrix} x_{ME} \\ y_{ME} \\ z_{ME} \end{bmatrix} = [L][C] \begin{bmatrix} R_m \cos\phi_{SRS} \cos\psi_{SRS} \\ R_m \cos\phi_{SRS} \sin\psi_{SRS} \\ R_m \sin\phi_{SRS} \end{bmatrix} \quad (1)
$$

where $\Phi_{SRS}$ and $\psi_{SRS}$ indicate the position of the Moon-based platform in the SRS; $\Phi_{SRS}$ is the Moon latitude and $\psi_{SRS}$ is the Moon longitude. $R_m$ indicates the radius of the Moon (we assumed the Moon is regularly spherical). [C] represents the transformation matrix from the ME system to the PA system. [L] represents the transformation matrix from the PA system to the SCRS system [41], which can be described by three Euler angles (Figure 2). In the left part of the formula, $(X_{SCRS}, Y_{SCRS}, Z_{SCRS})^T$ is the position of the Moon-based platform in the SCRS. In the right part of the formula, $(x_{PA}, y_{PA}, z_{PA})^T$ and $(x_{ME}, y_{ME}, z_{ME})^T$ are the position of the Moon-based platform in the PA system and ME system, respectively. The detailed description of the above rotation matrix can be found in Appendix A.

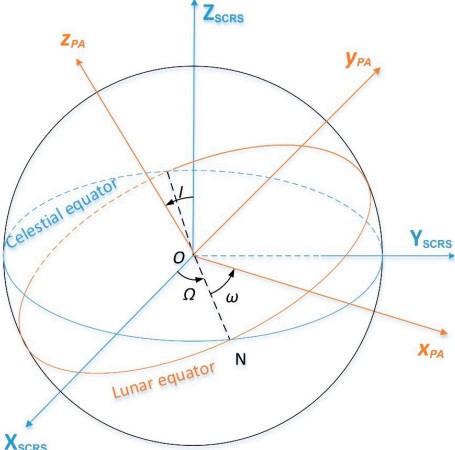

**Figure 2.** Angle rotation relationship between PA and SCRS.

### 2.2.2. Transformation from GRS to GCRS

There are also three steps to convert the GRS to the GCRS. The first step is to convert the GRS to ITRS. This step is slightly different from the conversion of the Moon's SRS to ME system. As we considered the Earth as an ellipsoid in this article, the second step was to convert the ITRS to the celestial intermediate reference system (CIRS). The CIRS is a geocentric reference system related to the GCRS by a time-dependent rotation that takes into account the precession–nutation [50]. Finally, the third step is to convert the CIRS to the GCRS. The transformation procedure is as follows:

$$
\begin{bmatrix} X_{GCRS} \\ Y_{GCRS} \\ Z_{GCRS} \end{bmatrix} = [Q] \begin{bmatrix} x'_{CIRS} \\ y'_{CIRS} \\ z'_{CIRS} \end{bmatrix} = [Q][\Theta][W] \begin{bmatrix} x_{ITRS} \\ y_{ITRS} \\ z_{ITRS} \end{bmatrix} = [Q][\Theta][W] \begin{bmatrix} (N+h)\cos\phi_{GRS}\cos\psi_{GRS} \\ (N+h)\cos\phi_{GRS}\sin\psi_{GRS} \\ \left((1-e^2)N+h\right)\sin\phi_{GRS} \end{bmatrix}, \quad (2)
$$

where

$$
N = \frac{a}{\sqrt{1 - e^2 \sin^2 \phi_{GRS}}}, \quad (3)
$$

Parameters $\Phi_{GRS}$ and $\psi_{GRS}$ indicate the position of the locations of the earth feature points in the GRS; $\psi_{GRS}$ is the Earth longitude and $\Phi_{GRS}$ is the Earth latitude. Parameter h indicates the altitude of the point, which was ignored in this research. Parameter e represents the ellipsoidal eccentricity of the Earth. Parameter a is the Earth equatorial radius. The rotation matrix [W] represents the polar motion matrix. $[\Theta]$ is the transformation matrix arising from the rotation of the Earth around the axis associated with the pole [50]. [Q] is the transformation matrix arising from the motion of the celestial pole in the celestial reference system. The detail of [W], $[\Theta]$, and [Q] are presented in Appendix A.

In the formula, $(X_{GCRS}, Y_{GCRS}, Z_{GCRS})^T$ and $(x_{ITRS}, y_{ITRS}, z_{ITRS})^T$ are the positions in the GCRS and ITRS. $(x'_{CIRS}, y'_{CIRS}, z'_{CIRS})^T$ is the position in the CIRS.

### 2.2.3. Transformation in the Celestial Reference System

So far, we have obtained the Moon-based platform position under the SCRS, the ground feature point positions under the GCRS, and the Sun position under the HCRS at time t. Next, we needed to convert the different reference systems to the geocentric celestial reference system. Ignoring the influence of relativity theory, the transformation between these systems is a simple conversion. Its expression is:

$$
\begin{bmatrix} X_{HCRS} \\ Y_{HCRS} \\ Z_{HCRS} \end{bmatrix} = \begin{bmatrix} X_{GCRS} \\ Y_{GCRS} \\ Z_{GCRS} \end{bmatrix} + \begin{bmatrix} X_E \\ Y_E \\ Z_E \end{bmatrix} = \begin{bmatrix} X_{SCRS} \\ Y_{SCRS} \\ Z_{SCRS} \end{bmatrix} + \begin{bmatrix} X_M \\ Y_M \\ Z_M \end{bmatrix}, \quad (4)
$$

where $(X_E,Y_E,Z_E)^T$ and $(X_M,Y_M,Z_M)^T$ are the coordinates of the Earth barycenter in the HCRS and Moon barycenter in the HCRS, respectively.

## 3. Moon-Based Observation Geometry

With the help of coordinate system transformation, some parameters regarding the observation characteristics can be calculated within the same reference system. We already know that placing sensors at different locations on the moon can have different effects on the imaging range and characteristics and observations from the same sensor to different earth feature points (EFP), which can also have a large difference in the coverage characteristics [10,36,40]. For the Arctic and Antarctic areas that are the focus of this paper, we simulated the long-term observation characteristics of the Moon-based platform at different latitudes in the polar circle. Through these characteristics, it can provide a reference for large-scale scientific observations of polar geoscience phenomena. The specific implementation is as follows: In order to evaluate the observation characteristics and compare the coverage differences between the Antarctic and Arctic, a lunar observatory site with good coverage should be selected. Therefore, we assumed that the lunar-based sensor was placed at the position of the near-side center of the Moon. We simulated the coverage duration, coverage frequency, and observation angles of the 12 feature points within the Antarctic and Arctic Circles on Earth. As a series of experiments have shown that the difference between the length of the observation duration and the distribution of the angle is only obvious with the latitude change, and the change is not obvious with the longitude change at the same latitude, we only discuss the observational characteristics of the feature points on the prime meridian. The principle of feature points selection is to set six feature points along the 0° warp within the Arctic and Antarctic Circles, respectively, where the latitude interval between each point is 5°. The location information of these points are as follows: 90°N 0°E, 85°N 0°E, 80°N 0°E, 75°N 0°E, 70°N 0°E, 65°N 0°E, 90°S 0°E, 85°S 0°E, 80°S 0°E, 75°S 0°E, 70°S 0°E, and 65°S 0°E. The simulated statistical time was from January 1, 1960, 00:00 (UTC) to January 1, 2060, 00:00 (UTC). Before the analysis, it is worth noting that for true observations, cloud cover and atmospheric refraction have a huge impact on the observation result, but the analysis in this study did not consider the impact of both.

In this section, we specifically analyze the observation characteristics of each feature point during 1960–2060. The first part was to calculate the position of the nadir points of the Moon and the Sun. The second part was to count the observation duration and observation frequency of each feature point, and eliminate the influence of the sunlight on the observation characteristics. In the third part, we calculated the observation angle and incident angle of each feature point, and summarized the characteristics of the angle combination. As a very important part of Moon-based geometry, the position of nadir points of the Moon and the Sun were calculated in previous studies [36,39], but it is the basis of the geometry study, and can provide some rules for the subsequent observation durations in this study. Therefore, this step is not listed in detail in this paper and the results are briefly explained here. For the position of the Moon's nadir points, the maximum latitude position can reach 18.5°~28.7° in each hemisphere, and in the same year, the maximum value is symmetrical in the Northern and Southern Hemispheres. The changing period of maximum latitude is about 18.6 years, showing a type of sinusoidal variation, and corresponding to the period of angular variation between the Moon's orbit and the Earth's equator. For the longitude of the nadir points, they exhibited a stable linear change between −180°−+180°. This means that a Moon-based sensor can cover the longitude uniformly. We also calculated the position of the Sun's nadir points. In 100 years, the latitude positions of the Sun's nadir points uniformly changed shape like a sinusoidal function between 23.45°N to 23.45°S. The change period is about 365.2422 days. The maximum latitude that can be reached by the Sun's nadir point is relatively stable and symmetrical in the Northern and Southern Hemispheres. The longitude positions of the Sun's nadir points showed a uniform variation throughout the study period, and the range of variation was always between −180°−+180°. In addition to the calculation of the nadir points, the other two parts are described in detail below.

### 3.1. Observation Duration and Frequency

As the important parameters of the Moon-based observation of the polar region, the observation duration and frequency can reflect its performance. A simplified model of Moon-based Earth observations is shown in Figure 3. As shown, the blue region is the region covered by the Moon-based observations, the yellow region is the region that the Sun can illuminate, and the overlapping portion is the observable region of the Moon-based optical sensor that only considers optical observation. As we approximate sunlight as parallel light, sunlight can reach the entire hemisphere of the Earth, and we know that there is an Earth viewing angle of about one degree of the Moon-based observation [39], so the observation area is slightly smaller than the hemisphere. In Figure 3, the red area at the top is the boundary of the Arctic region, and the purple area at the bottom is the boundary of the Antarctic region; point T $(X_T, Y_T, Z_T)$ is the location of the feature point on the surface; $O_e$ $(X_e, Y_e, Z_e)$ is the barycenter of the Earth; $O_s$ $(X_s, Y_s, Z_s)$ and $O_m$ $(X_m, Y_m, Z_m)$ are the position of the solar barycenter and the position of the Moon-based observation platform, respectively; and $\alpha$ and $\beta$ are the observation zenith angle and the solar incident angle, respectively. Without considering the refraction of light and the optical properties of the sensor, theoretically, when $\alpha$ is less than 90 degrees, the EFP are within the range of the Moon's line of sight. When $\beta$ is less than 90 degrees, the EFP are in the solar illumination range, and when $\alpha$ and $\beta$ are simultaneously less than 90 degrees, the EFP are within the observation range of the Moon-based optical sensor. The formulas for calculating $\alpha$ and $\beta$ are shown in Appendix A.

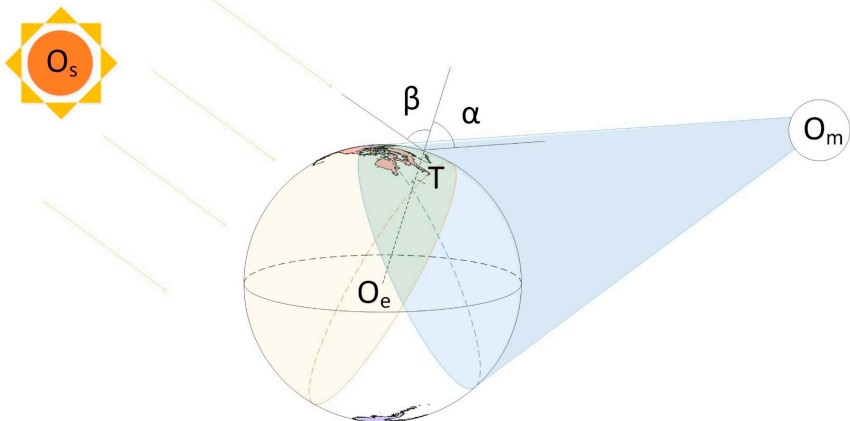

**Figure 3.** Simplified model of Moon-based Earth observations where Os, Oe, and Om are the barycenter of the Sun, Earth, and Moon; T is the location of the EFP; and $\alpha$ and $\beta$ are the observation zenith angle and the solar incident angle.

Using the formula of angles $\alpha$ and $\beta$, we could directly calculate the visibility of the special moments of each EFP, but the observation duration corresponding to the length of the observable time interval needed to be further calculated. Therefore, each observation moment was assumed to represent a one-second observation duration and continuous uninterrupted observations were recorded as a frequency of one. Using the above scheme, we calculated the total coverage time, the maximum (continuously) coverage duration, and the access frequency of each point in the 100 year period, and then calculated the average coverage time. The comparison results are shown in Figures 4 and 5. Figure 4 shows the access duration and frequency of the Moon-based platform to the EFP in the Earth's Arctic region where the dark blue represents the coverage of the Moon-based platform without considering the Sun, and light blue represents the coverage of the Moon-based platform considering the illumination. In both cases (dark blue and light blue), the total coverage time of the Earth's Arctic region remain stable with the latitude change, but the latter one (light blue) shortened the total time by considering the lighting. As the latitude decreased, the single maximum coverage duration in both cases showed a downward trend, especially for the case that considered sun exposure, and the maximum coverage duration exhibited a slow decline first and then a rapid decline after. It is worth

noting that when the latitude was higher than 70°, the difference in illumination or not illumination was not that obvious. However, when the latitude was below 70°, the maximum coverage duration considering the illumination was significantly attenuated. Regardless of the solar illumination, the single maximum coverage duration was reduced from 347.4 hours (about 14.5 days) at the North Pole to 121.8 hours (about 5.1 days) at 65°N. Considering solar illumination, the single maximum coverage duration was reduced from 347.0 hours (about 14.5 days) at the North Pole to 20.9 hours (about 0.87 days) at 65°N. As the latitude decreased, the access frequency increased gradually in both cases, and the average coverage time first showed a rapid drop, then slowly declined. Figure 5 shows that the coverage characteristics changed in the Antarctic region. In this figure, the dark orange represents the coverage of the Moon-based platform without considering the Sun, and light orange represents the coverage of the Moon-based platform considering the illumination. Compared with Figure 4, it was found that the parameters of the bipolar EFP coverage characteristics showed similar trends. Regardless of the solar illumination, the single maximum coverage duration was reduced from 347.2 hours (about 14.5 days) at the South Pole to 121.6 hours (about 5.1 days) at 65°S. Considering solar illumination, the single maximum coverage duration was reduced from 347.2 hours (about 14.5 days) at the South Pole to 20.9 hours (about 0.87 days) at 65°S. It can be concluded that in the polar region, the frequency of access increased uniformly with the decrease in latitude, while the single coverage duration decreased, the total coverage time of each latitude only slightly fluctuated, and the trends in the Arctic and Antarctic were the same. In the summer of the Northern and Southern Hemispheres, optical observations of up to 14.5 days could be achieved at pole points, and continuous optical observations of up to 0.87 days could be achieved even when the latitude was reduced to 65°N or 65°S. This feature can well serve the field of geoscience in the large-scale, long-term continuous change monitoring in the polar regions.

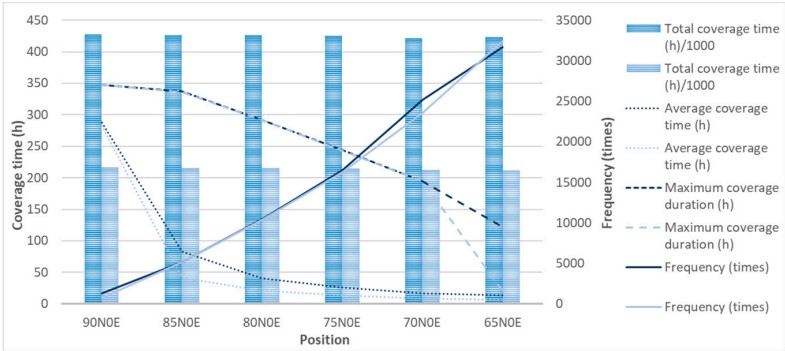

**Figure 4.** The duration and frequency of the Moon-based platform to the EFP in the Arctic region.

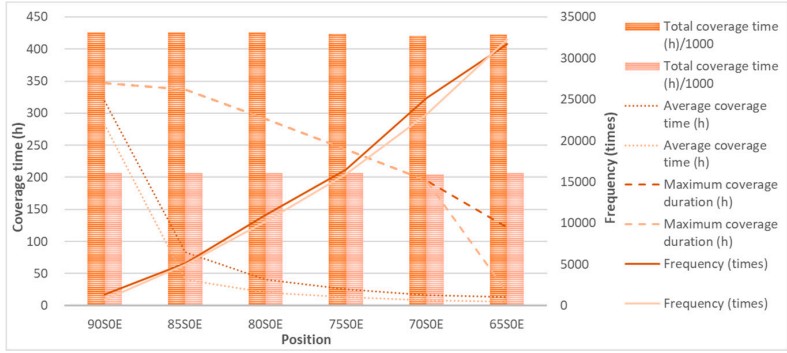

**Figure 5.** The duration and frequency of the Moon-based platform to the EFP in the Antarctic region.

The annual coverage time and access frequency of each EFP were counted separately, and the annual solar coverage time and frequency of each EFP were counted for comparison. The results are shown in Figures 6 and 7. From Figure 6, it can be seen that the variation of the sunshine time in

each year was not obvious, and the time changes with the latitude was also very short. However, the frequency of sunshine increased with the decrease in latitude from 1–3 times in the pole points every year to 365 times in 65°N or 65°S, and the fluctuations in the frequency of sunshine exposure at each latitude were not more than three times. Comparing the sunshine time and frequency at the same latitude in the Northern and Southern Hemispheres, we found that the sunshine time in the southern hemisphere was slightly shorter than the Northern Hemisphere, but the frequency was one time more than that in the Northern Hemisphere. This is mainly because we artificially set January 1 as the inter-annual boundary, while the Southern Hemisphere polar region was now in the continuous sunshine period.

The annual coverage time statistical results of each EFP at different latitudes are shown in Figure 7. It can be seen from the figure that the observation time exhibited a regular sinusoidal change with the year, which has an interval of about nine years, and the variation range was about $7 \times 10^6 - 8.6 \times 10^6$ s (the total sunshine time was about $1.6 \times 10^7$ s). Comparing the same latitudes in the Northern and Southern Hemispheres, it can be seen that the direction of the wave was the opposite. As the time of sunshine does not change significantly between years, it can be concluded that the change is caused by the regular activity between the Moon and the Earth. Consistent with the total observation time at different latitudes, the annual coverage time did not change significantly with the change of latitude.

From the figure, we also noticed the access frequency of each EFP. Comparing Figure 7 with Figure 6, the fluctuation amplitude of the Moon-based observation access frequency was much larger than the fluctuation amplitude of the solar radiation frequency. The fluctuation range of the observation access frequency increased from about 7–9 times in the pole points to 270–360 times in 65°N or 65°S. The observation frequency showed a significant increase with the decrease in latitude, which is directly related to the increase in sunshine frequency with the latitude reduction. However, there is another feature of frequency change in the high latitudes, where the interannual variation of the access frequency was not obvious, but as the latitude decreased, the wave dynamic potential gradually appeared. This fluctuation showed a change period of 18.6 years, which was consistent with the fluctuation frequency of the nadir points of the Moon. This is because every 18.6 years, the angle between the orbit of the Moon plane and the Earth equator plane reaches a maximum of 28.65° [38].

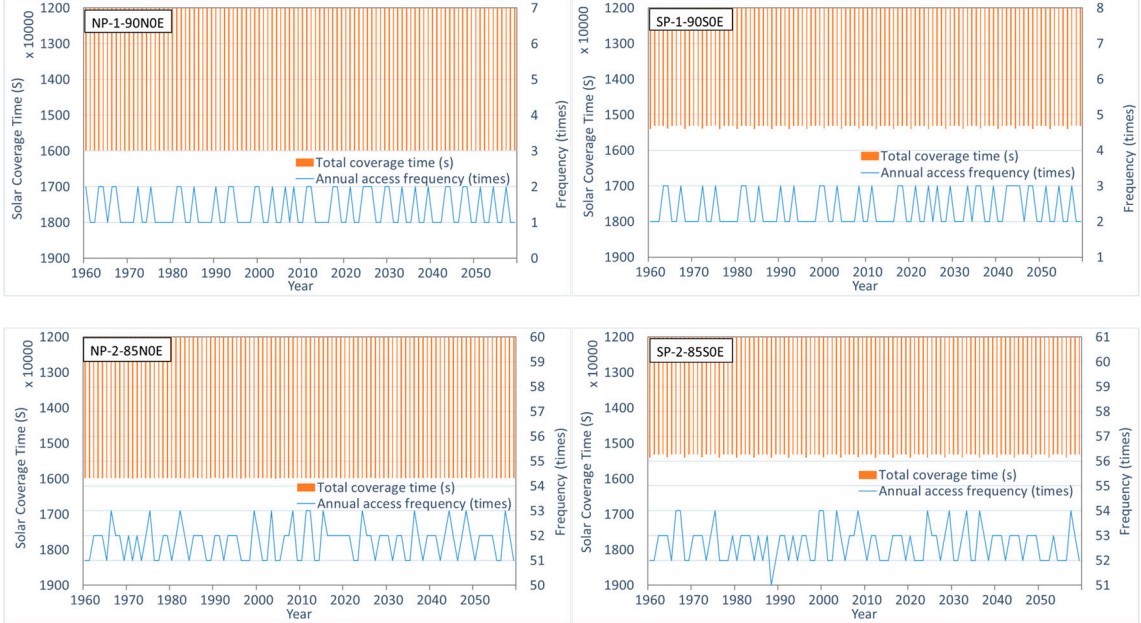

**Figure 6.** *Cont.*

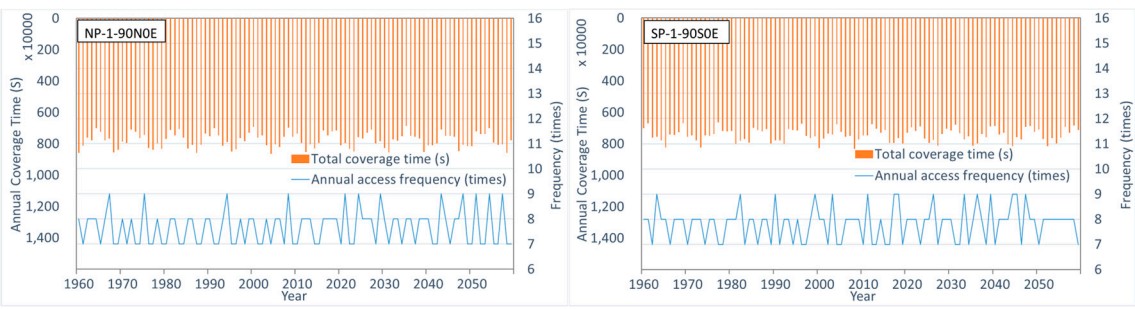

**Figure 6.** The annual solar coverage time and frequency of each EFP. (**Left**) Arctic; (**right**) Antarctic.

**Figure 7.** *Cont.*

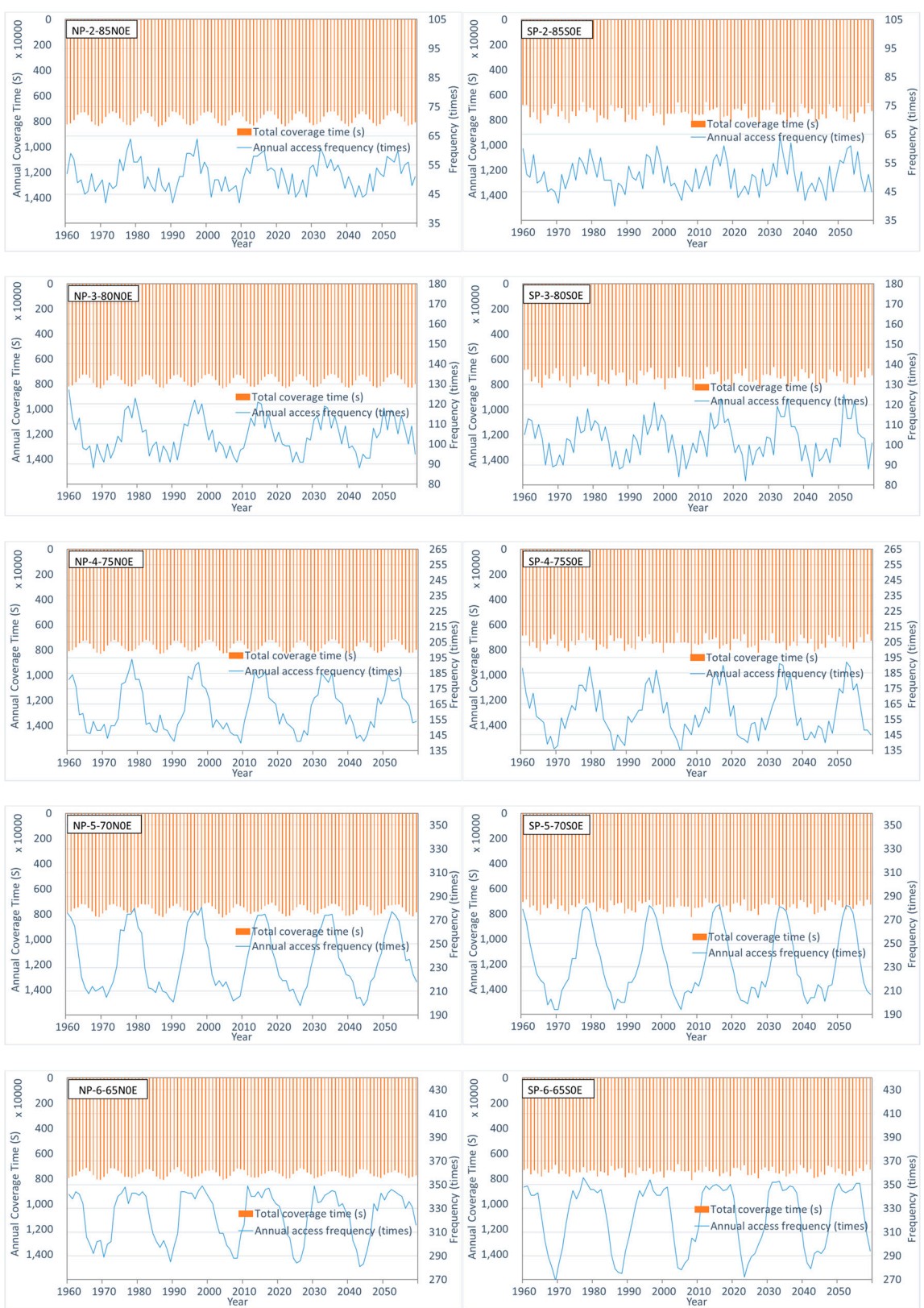

**Figure 7.** The annual coverage time and access frequency of each EFP. (**Left**) Arctic; (**right**) Antarctic.

The detail of the annual coverage time range for different latitudes is shown in Figure 8. We made the duration of 366 days as the x-axis and the total study period of 1960–2060 as the y-axis, and plotted the coverage duration and frequency characteristics of the annual Moon-based optical observations.

The blue rectangle represents the observable time by the Moon-based optical sensor, the white area represents the unobservable time by the Moon-based optical sensor, and the annual result occupies a row in the chart, which accumulates year by year. As can be seen from the figure, the dominant observation period at the North Pole was concentrated from the 80th day to the 265th day of the year (January 1 was considered the beginning of the year), that is, from mid-March to mid-September. During this period, the distribution of observation duration showed regular changes within one year and between different years. Within one year, the observable period and the unobservable period alternately appeared. Between different years, the observable period had a short overlap between the adjacent two-years. As the latitude decreased, the entire observable period extended toward both boundaries. For the point of 85°N0°E, the dominant observation period was concentrated from the 67th to the 277th day of the year; for the point of 80°N0°E, the observation period was concentrated from the 54th to the 290th day of the year; for 75°N0°E, it expanded from the 39th to the 305th day of the year; for 70°N0°E, it expanded from the 21th to the 323th day of the year; and for 65°N0°E, the observable period had extended to the whole year. As the latitude decreased, the length of a single observation duration shortened and the frequency of observation increased. This shortening was more significant at the boundary of the entire observation interval. The single continuous observation duration could reach up to 13–14 days at the North Pole, however, when the latitude dropped to 65°N, the length of a single observation duration fell to less than one day. The dominant observation period at the South Pole was concentrated from the first day to the 78th day and the 266th day to the 365th day of the year, that is, from mid-September to the following year's mid-March. During this period, the distribution of the observation duration showed the same regular changes as the Arctic. As the latitude decreased, the entire observable period extended toward both boundaries. For the point of 85°S0°E, the dominant observation period was concentrated from the first to the 90th day and the 253th to the 365th day of the year; for the point of 80°S0°E, the observation period was concentrated from the first to the 104th day and the 239th to the 365th day of the year; for 75°S0°E, it expanded from the first to the 119th day and the 224th to the 365th day of the year; for 70°N0°E, it expanded from the first to the 138th day and the 205th to the 365th day of the year; for 65°S0°E, the observable period extended to the whole year. With the decrease in latitude, the length of a single observation duration decreased and the frequency of annual visits increased, showing a trend consistent with the Northern Hemisphere. The length of a single observation duration dropped from 13–14 days at the pole to less than a day at 65°S.

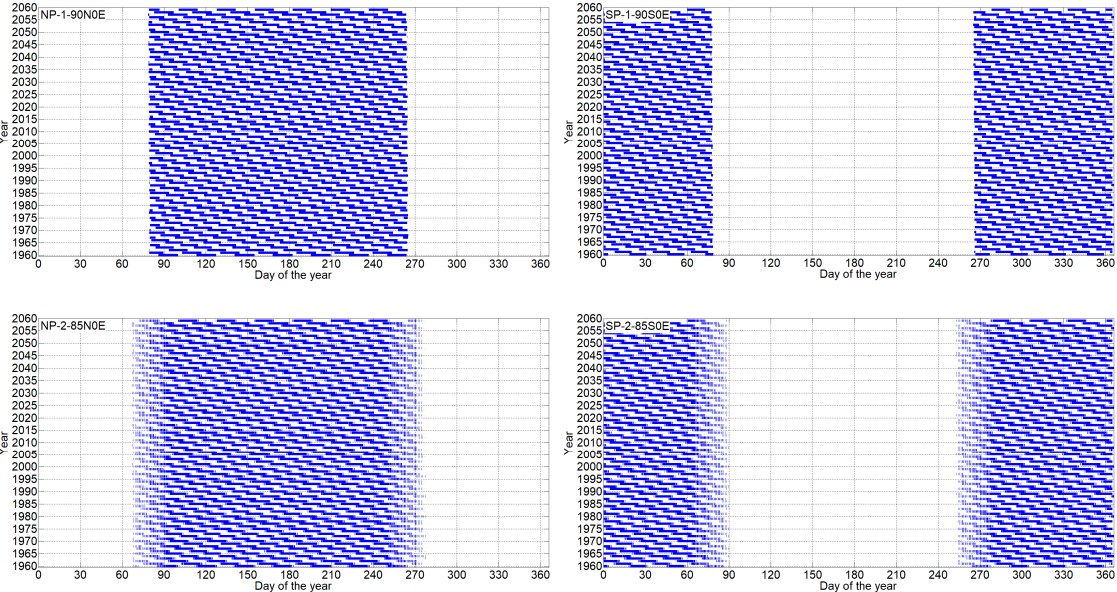

**Figure 8.** *Cont.*

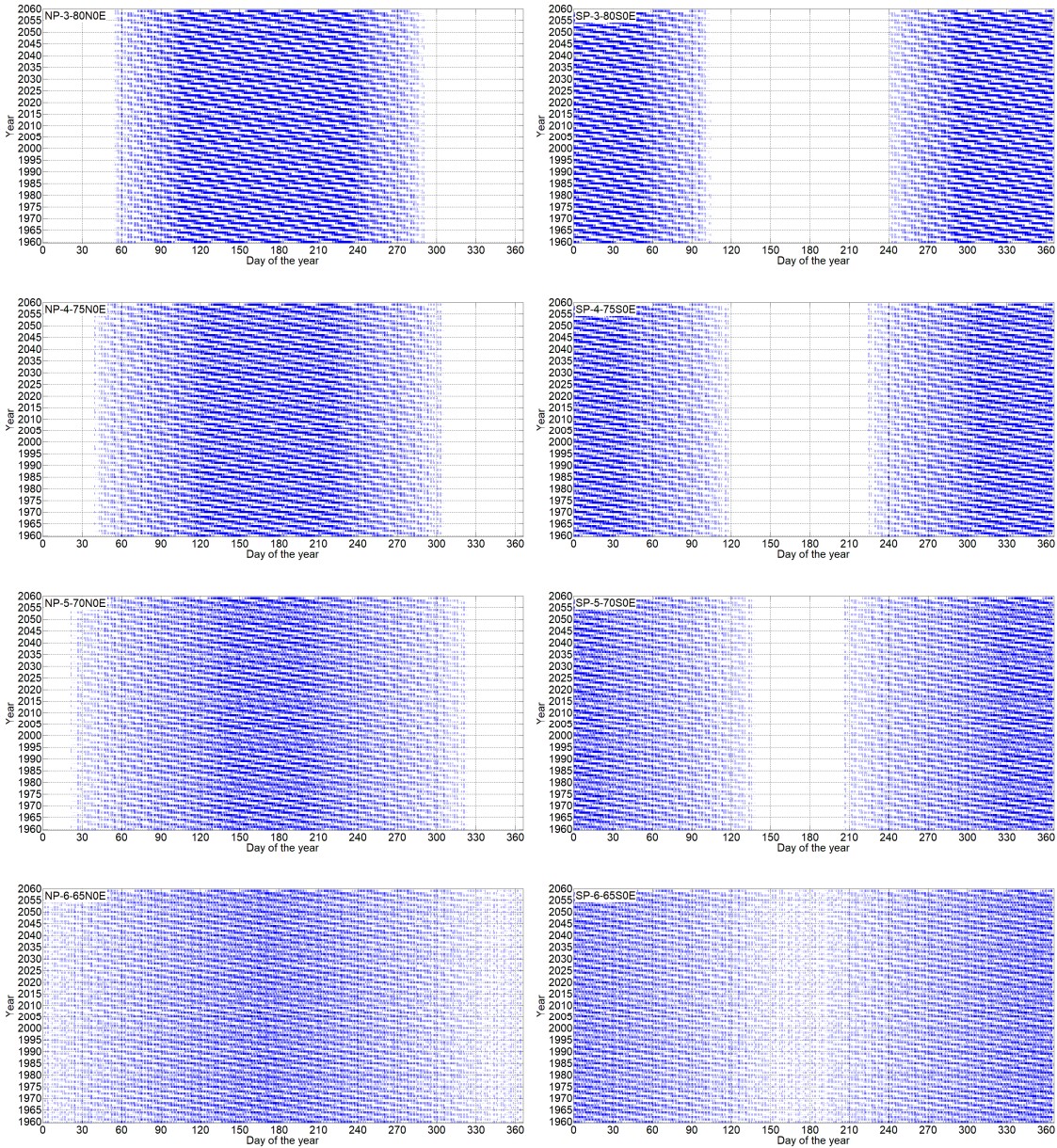

**Figure 8.** Detail of the annual coverage time range of each EFP. (**Left**) Arctic; (**right**) Antarctic.

*3.2. The Change of Solar Incident Angle and Observation Angle*

Angle information is an important parameter that cannot be ignored in optical remote sensing. With the development of quantitative remote sensing, scientists are increasingly recognizing the diversification and precision of applications brought by multi-angle remote sensing. The typical angle parameters we referred to in this paper are the zenith, azimuth, and elevation. The relationship between these three angles is as follows:

(1)   Zenith Angle: Centered on the surface of the Earth, the opposite direction of gravity is the zenith direction. The angle between the light and the zenith is called the zenith angle.

(2)   Elevation Angle: Through the Earth surface point, we made a plane that is perpendicular to the zenith direction, which is called the horizontal plane, and the angle between the light and the horizontal plane is called the elevation angle, which is obviously the complementary angle of the zenith angle [51].

(3)   Azimuth Angle: When the light is projected onto the horizontal plane, the angle between the projection direction and the true north direction here is called the azimuth angle [37].

For the surface EFP, we considered that two rays were very important. One is the Sun's incident light, and the other is the Sun's reflected light, which is emitted to the Moon-based observatory, and is also the direction of sight. For these two rays, they respectively correspond to the three angle parameters above-mentioned. These are the view zenith angle (VZA); the solar zenith angle (SZA); view elevation angle (VEA); solar elevation angle (SEA); view azimuth angle (VAA); and solar azimuth angle (SAA). Variations and combinations of these angles can have a direct impact on the observations. Therefore, we performed a long-term sequence simulation study on the angular information of all feature points. As the zenith angle and the elevation angle complement each other, in this paper, we only calculated the zenith angle and azimuth angle. The formula for the observation of the zenith angle and the solar incident angle (solar zenith angle) was introduced in Section 3.1. In order to calculate the VAA and SAA information, we introduced a new reference system, the horizontal coordinate system (HCS), also known as the topocentric coordinate system that used the target's local horizon as the fundamental plane [10]. Thus, the target was selected as the coordinate origin, the north direction is the $y_h$-axis and the zenith direction of the target is the $z_h$-axis. The $x_h$-axis follows the right-hand rule and points east of the target. Azimuth is the angle of the object around the horizon, usually measured from true north and increasing eastward. Then, the horizontal coordinate system is expressed as $[x_h, y_h, z_h]^T$. In order to realize the transformation from ITRS to HCS, we needed to separate it into two steps: translation and rotation. Translation is to move the origin from the Earth's center to the Earth's surface. We have already previously introduced the translation of the coordinate system. Rotation is the conversion of two coordinate systems by rotating a certain angle. Two parameters are used to determine the rotation transformation: the rotation axis and the rotation angle. In this situation, the coordinates are rotated around the Z-axis first, and then rotated around the X-axis. Furthermore, the rotation angle is determined by the position of the point ($\Phi_{GRS}$, $\psi_{GRS}$). In short, the rotation process can be simply summarized as rotating $\psi_{GRS}$ +90 degrees counterclockwise around the Z-axis and then rotating 90-$\Phi_{GRS}$ degrees clockwise around the X-axis. The transformation formula is as follows:

$$
\begin{bmatrix} X_{HCS} \\ Y_{HCS} \\ Z_{HCS} \end{bmatrix} = R_x(90° - \Phi_{GRS})R_z(-(\psi_{GRS} + 90°))\left( \begin{bmatrix} x_{ITRS} \\ y_{ITRS} \\ z_{ITRS} \end{bmatrix} - \begin{bmatrix} x_p \\ y_p \\ z_p \end{bmatrix} \right), \tag{5}
$$

$$
\begin{bmatrix} X_{HCS} \\ Y_{HCS} \\ Z_{HCS} \end{bmatrix} = \begin{bmatrix} -\sin\psi_{GRS} & -\cos\psi_{GRS} & 0 \\ \sin\Phi_{GRS}\cos\psi_{GRS} & -\sin\Phi_{GRS}\sin\psi_{GRS} & \cos\Phi_{GRS} \\ -\cos\Phi_{GRS}\cos\psi_{GRS} & \sin\psi_{GRS}\cos\Phi_{GRS} & \sin\Phi_{GRS} \end{bmatrix}\left( \begin{bmatrix} x_{ITRS} \\ y_{ITRS} \\ z_{ITRS} \end{bmatrix} - \begin{bmatrix} x_p \\ y_p \\ z_p \end{bmatrix} \right), \tag{6}
$$

where $[X_p, Y_p, Z_p]^T$ represents the position of the feature point ($\Phi_{GRS}$, $\psi_{GRS}$) in ITRS, and it is also the origin of HCS. Taking advantage of the rotation matrix, the coordinates are transformed to the $X_{HCS}$, $Y_{HCS}$, and $Z_{HCS}$ frame. Thus, the solar and view azimuth angles of 12 earth feature points are calculated based on the trigonometric function formula.

In order to better reveal the angle characteristics, we combined the elevation and azimuth angles of each latitude on a polar plot. In the diagram, the polar angle corresponds to the azimuth angle and the radius corresponds to the elevation angle. We simulated the angles that can be observed by the Moon-based optical sensors for all feature points over a 100-year period (calculation frequency: once per hour). The VEA/VAA of Moon-based Earth observations and the SEA/SAA of solar incident light at a specific point in the central meridian of the Northern and Southern Hemispheres are all summarized in Figure 9. Dark blue represents the VEA/VAA in the Northern Hemisphere (90°N, 85°N, 80°N, 75°N, 70°N, 65°N); light blue represents the VEA/VAA in the Southern Hemisphere (90°S, 85°S, 80°S, 75°S, 70°S, 65°S); dark orange represents the SEA/SAA in the Northern Hemisphere (90°N, 85°N, 80°N, 75°N, 70°N, 65°N); and light orange represents the SEA/SAA in the Southern Hemisphere (90°S, 85°S, 80°S, 75°S, 70°S, 65°S).

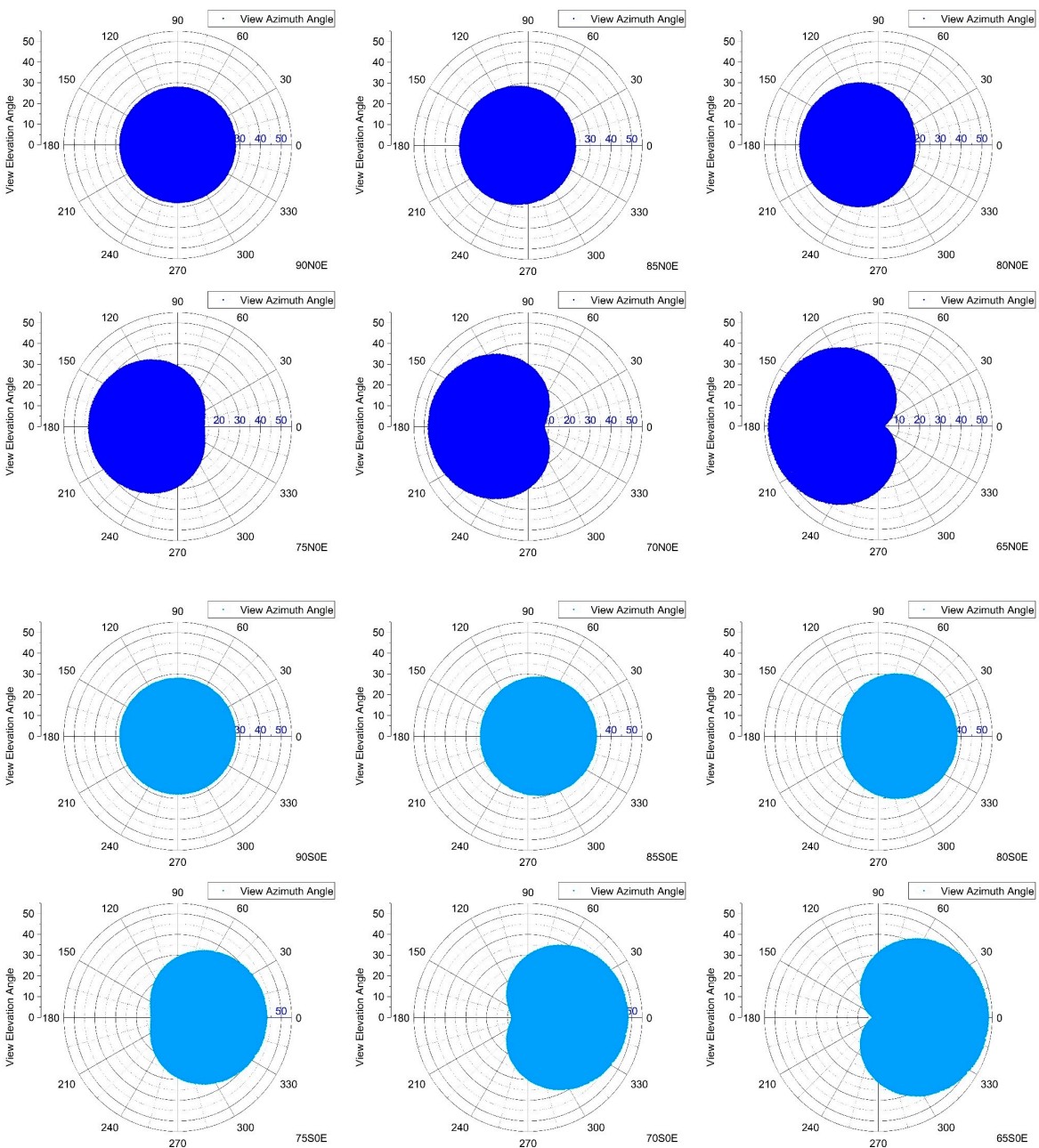

**Figure 9.** *Cont.*

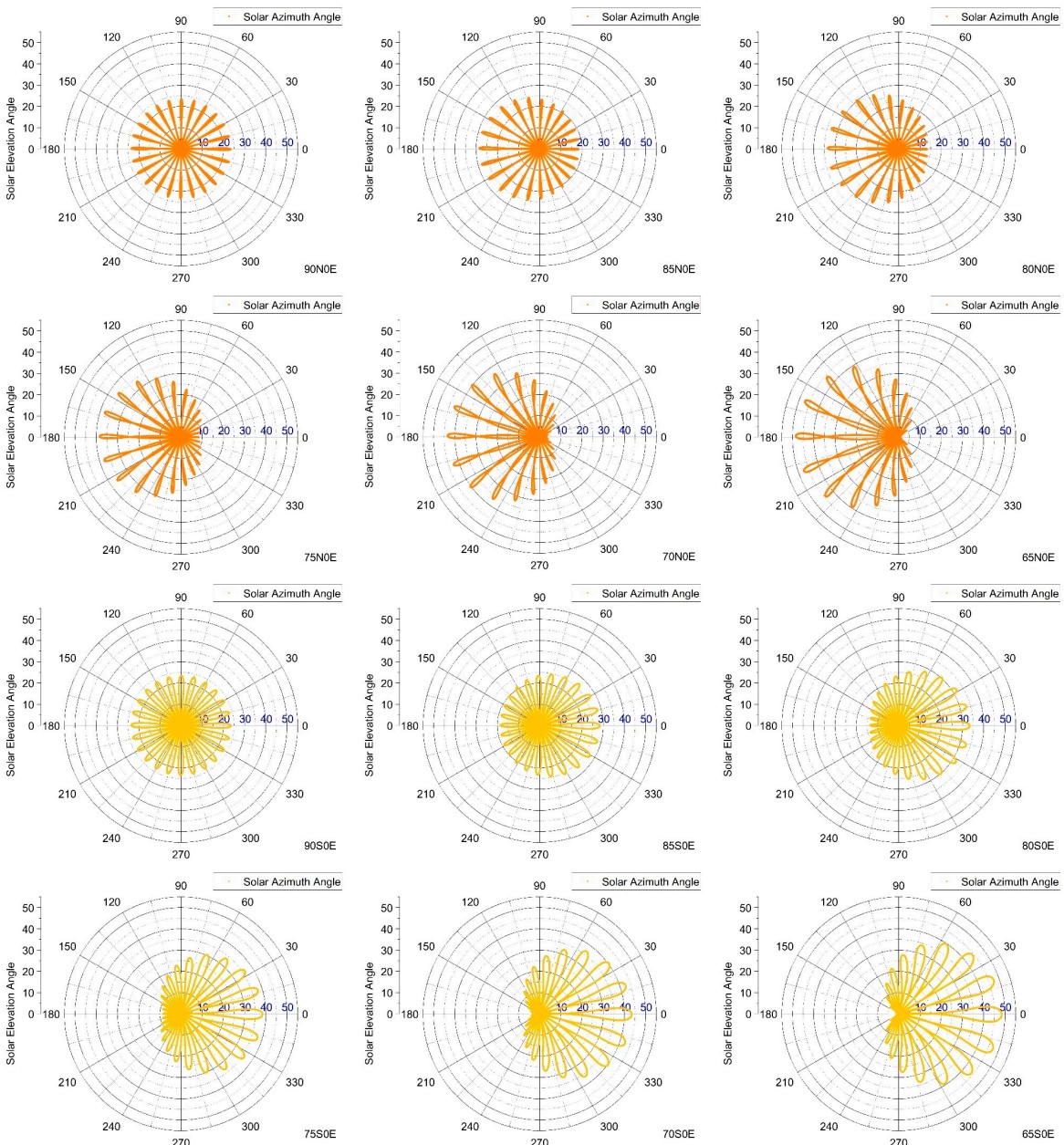

**Figure 9.** Azimuth–elevation angle combination distribution of Moon-based Earth observations and solar incidence during 1960–2060.

For the VEA/VAA of Moon-based observations, the VAA of the feature points 90N0E and 90S0E could uniformly cover the complete 0°–360° interval, and the VEA could cover the 0°–28° interval. The combined result exhibited a symmetric characteristic. The VAA of the feature points 85N0E and 85S0E could cover the 0°–360°, and the VEA could cover the 0°–33° interval, but the azimuth–elevation combination exhibited a southward (85N0E) offset or northward (85S0E) offset, respectively. The angle combination was only symmetrical in the east–west directions, and the corresponding offset situation occurred in the north–south directions. The feature points 80N0E and 80S0E, 75N0E and 75S0E, and 70N0E and 70S0E continued to exhibit the above-mentioned angular combination characteristics. The angle combination was gradually southward offset with the decrease in latitude in the Northern Hemisphere, and the angular combination showed a gradually increasing absence in the northern direction. The maximum VEA showed an increasing trend in a step-length of 5°. At the feature points 65N0E and 65S0E, the angular combination continued to maintain the above characteristics.

The maximum VEA could be as high as about 53°, while the VAA was concentrated in the south (65N0E) or north direction (65S0E), and the angular combination showed an absence in the north (65N0E) or south (65S0E) directions, respectively.

For the SEA/SAA of the sun's incident light, the SEA of the feature points 90N0E and 90S0E presented a petal shape. This is because we chose a time interval of one hour, and the space between the petals indicated the unselected time point. This indicates that if we choose a dense interval, the solar azimuth–elevation combination can achieve a full coverage of 0–360°. For the pole points, the SEA varied from 0–24°, showing symmetrical characteristics. Additionally, the figure showed a full (0–360°) SAA distribution. At the feature points 85N0E and 85S0E, the azimuth–elevation combination exhibited a southward offset or northward offset, respectively, and its direction was consistent with the angle combination change of the Moon-based observation angle. The SEA increased by 5° on the basis of the pole points, reaching 29°. Continuing to reduce the latitude, presenting a similar movement characteristic of the Moon-based observation angle combination, and the SEA gradually increased by 5°. At the feature points 65N0E and 65S0E, the angle combinations had a large number of absences in the north and south, respectively, and the SEA varied from 0 to 49°.

From the above analysis, it can be concluded that the Moon-based platform can have sufficient observation angles to the polar regions. With the development of quantitative remote sensing, angular information and characteristics have received attention due to their huge influence on observations. The zenith and azimuth angles directly determine the accuracy of the observed results [37]. The relative azimuth and zenith angles can affect the calculation of BRDF, and provide a basis for inversion of the surface albedo [52]. The albedo is an important parameter for further research on radiant energy balance. Furthermore, the angle information must be considered when interpreting thermal radiances observed over land surfaces, which is also an important parameter for energy budgeting research [45]. Based on the above background, we can conclude that the sufficient angle information of the Moon-based platform can provide strong support for albedo inversion, Earth energy balance, and atmospheric directional radiation, which can help us to deepen our understanding of Earth surface activities.

## 4. Discussion

The previous sections introduced the observation duration, frequency, and angles of the Moon-based Earth observation platform. Our study suggests that mid-March to mid-September is the dominant observation time in the Northern Hemisphere. For the Southern Hemisphere, the dominant observation time occurs from mid-September to mid-March of the following year. The maximum single observation time of both poles can reach up to 14.5 days. In addition, sufficient angle combination features covered most of the land and sea area of the North and South Poles.

Considering the characteristics of the Moon-based observation on the Earth's polar regions, we have discovered some unique geoscience phenomena suitable for lunar observations in the polar regions of the Earth. By calculating the annual average sea ice area, it can be seen that from 1989 to 2018, the Arctic sea ice area showed a significant decreasing trend, and the Antarctic sea ice area showed a slight upward trend before 2014, but has decreased significantly in the past four years (Figure 10). By calculating the changes in sea ice area during the year, it can be seen that although the sea ice area has different variations during the interannual of the 30 years, the freezing and thawing times of the sea ice in the North and South Poles remained basically unchanged (Figure 11). The thawing time of Arctic sea ice begins around the 60th day of the year and the freezing time begins around the 255th day; the freezing time of the Antarctic sea ice begins around the 50th day of the year, and the thawing time is about the 270th day of the year. Due to the calculation of the observation characteristic of the Moon-based platform, we know that it has a period of dominant observation time from March to September near the North Pole, and has a period of dominant observation time from September to March of the following year near the South Pole. This feature facilitates direct observation of the ablation of sea ice in the Arctic and Antarctic, combined with sufficient angular characteristics, which can provide sufficient observational assurance for determining the correlation between global radiant

energy balance and sea ice area changes. The specific implementation of this research will be carried out in the future.

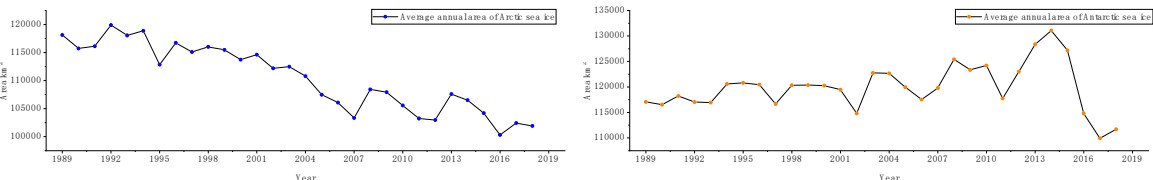

**Figure 10.** Interannual variation of sea ice area in the Arctic (**left**) and Antarctic (**right**) region from 1989 to 2018 (data were downloaded from the U.S. National Snow and Ice Data Center website [53]).

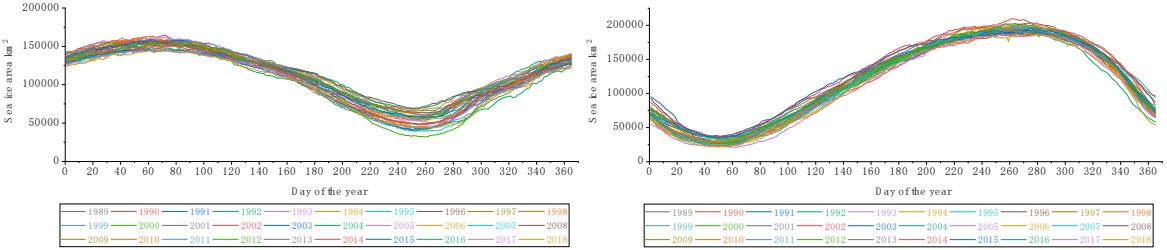

**Figure 11.** Sea ice area changes within the year in the Arctic (**left**) and Antarctic (**right**) region (data were downloaded from the U.S. National Snow and Ice Data Center website [53]).

## 5. Conclusions

In this article, we used the ephemeris DE430 and Earth orientation parameters as input data to derive the position and attitude of the Sun, Earth, and Moon. Then, we used the coordinate system transformation method to unify the position of the Moon-based platform and the location of the Earth's feature points into a unified coordinate system, and realized the simulation of ground objects observation from the perspective of the Moon. This article mainly analyzed the position of nadir points of the Sun and the Moon, the observation duration and frequency of the Moon-based observation, and the characteristics of the angles related to observation during the whole study period. The observation characteristics of feature points at different locations in the Antarctic and Arctic regions are summarized below.

For the entire study period, the total coverage time of observations at each latitude was almost constant. We found that the closer to the pole, the easier it was to obtain a larger continuous observation time. For the Arctic and Antarctic poles, a continuous observation time of 14.5 days could be obtained, but at the same time, the annual observation frequency was reduced to about ten times per year. In contrast, the closer we approached the equator, the more we could obtain the observation frequency. At the latitude of 65° in the Northern and Southern Hemispheres, the observation frequency could reach about 350 times per year, almost close to once a day. However, for these areas, we could only obtain a maximum continuous observation duration of less than a day. The study also showed that mid-March to mid-September is the dominant observation time in the Northern Hemisphere, and for the Southern Hemisphere, the dominant observation time occurs from mid-September to March of the following year.

For the annual coverage time and frequency of each feature point, the annual coverage time does not change with the latitude change, but there was a fluctuation in the cycle of approximately nine years during 1960–2060. The observed frequency increased with the decreasing latitude, and gradually appeared as regular fluctuations with a period of 18.6 years. Comparing the stable coverage time and frequency characteristics of the sunlight to each feature point, it was preliminarily judged that the above regular changes were caused by the relative position changes of the Moon and the Earth.

For the polar regions, we know it has a period of a dominant observation time. The observation period at the North Pole is concentrated from the 80[th] day to the 265[th] day of the year, and at the South

Pole, it is concentrated from the first day to the 78$^{th}$ day and the 266$^{th}$ day to the 365$^{th}$ day of the year. During this period of time, the distribution of observation duration showed regular changes within one year and between different years. As the latitude decreased, the entire observable period extended toward both boundaries, until for the points 65°N0°E and 65°S0°E, the observable period extended to the whole year. The length of a single observation duration shortened and the frequency of observations increased. The single continuous observation duration could reach up to 13–14 days at the North and South Poles. When the latitude dropped to 65°N or 65°S, the length of a single observation duration fell to less than a day. This feature of the Moon-based Earth observations contributes to a wide range and continuous observations of geosciences in the polar regions.

This study summarizes the observation angle characteristics of the feature points in the polar circle. For the VEA/VAA of Moon-based observations, the VAA of the pole points could uniformly cover the complete 0°–360°, and the VEA could cover the 0°–28° interval. Then, for every 5° reduction in latitude, the range of VEA increased by 5°. The angular combination distribution gradually shifted southward with the latitude in the Northern Hemisphere, and gradually shifted northward with the latitude in the Southern Hemisphere. For the SEA/SAA of the Sun's incident light, the angular combination distribution exhibited a variation law consistent with the observation angle of the Moon-based platform. At the pole, the SAA could cover the complete 0°–360°, the SEA range was 0–24 degrees, and as the latitude decreased, the maximum SEA increased by the corresponding degree.

The above experimental results show that the Moon-based platform can carry out long-term and large-scale observations of the North and South Poles of the Earth, and has the ability to obtain rich observations and incident angle information, which can be applied to the field of Earth observation as a new type of remote sensing technology.

**Author Contributions:** Conceptualization, G.L., Y.S., and Y.R.; Formal analysis, Y.S.; Funding acquisition, H.G. and G.L.; Investigation, Y.S.; Methodology, Y.S., G.L., and Y.R.; Project administration, H.G. and G.L.; Software, Y.S.; Supervision, H.G. and G.L.; Writing–original draft, Y.S.; Writing–review & editing, Y.S., H.G., G.L., and Y.R.

**Funding:**    This research was supported by the Key Research Program of Frontier Sciences CAS (QYZDY-SSW-DQC026) and the National Natural Science Foundation of China (41590852, 41001264).

**Conflicts of Interest:** The authors declare no conflicts of interest.

**Appendix A**

Detailed descriptions of rotation matrices C and L are expressed below.

The transformation matrix from the ME system to the PA system can be expressed as:

$$[C] = R_z(C_1)R_y(C_2)R_x(C_3), \tag{A1}$$

where C1, C2, and C3 are the transition angles, which are all constants but change with the JPL Ephemeris. Rx, Ry, and Rz are right-handed rotations of frame orientations, and the matrices mentioned in this paper all follow the following principles:

$$
\begin{aligned}
R_x(\alpha) &= \begin{pmatrix} 1 & 0 & 0 \\ 0 & \cos\alpha & \sin\alpha \\ 0 & -\sin\alpha & \cos\alpha \end{pmatrix}, \\
R_y(\alpha) &= \begin{pmatrix} \cos\alpha & 0 & -\sin\alpha \\ 0 & 1 & 0 \\ \sin\alpha & 0 & \cos\alpha \end{pmatrix}, \\
R_z(\alpha) &= \begin{pmatrix} \cos\alpha & \sin\alpha & 0 \\ -\sin\alpha & \cos\alpha & 0 \\ 0 & 0 & 1 \end{pmatrix},
\end{aligned}
\tag{A2}
$$

The Euler angle matrices can be given by:

$$[L] = R_z(-\omega)R_x(-I)R_z(-\Omega),$$ (A3)

Among them, the three Euler angles $\omega$, I, $\Omega$ can be found in the ephemeris file.

Expression for the transformation matrix [W] for polar motion is as follows:

$$[W] = R_z(-s\prime)R_y(x_p)R_x(y_p),$$ (A4)

where $x_p$ and $y_p$ are the "polar coordinates" of the celestial intermediate pole (CIP) in the ITRS and s′ is a quantity, named the "terrestrial intermediate origin locator", which provides the position of the TIO on the equator of the CIP corresponding to the kinematical definition of the non-rotating origin (NRO) in the ITRS, when the CIP is moving with respect to the ITRS due to polar motion. The expression of s′ as a function of the coordinates $x_p$ and $y_p$ is:

$$s\prime(t) = \frac{1}{2} \int_{t_0}^{t} (x_p \dot{y}_p - \dot{x}_p y_p)dt,$$ (A5)

Expression for the Earth rotation transformation matrix [$\Theta$] is:

$$[\Theta] = R_z(-ERA),$$ (A6)

where ERA is the Earth rotation angle between the celestial intermediate origin (CIO) and the TIO at date t on the equator of the CIP.

The transformation matrix for the celestial motion [Q] can be expressed as:

$$[Q] = R_z(-E)R_y(-d)R_z(E)R_z(s),$$ (A7)

where E and d help to define the coordinates of the CIP in the GCRS as follows:

$$X = \sin d \cos E, \ Y = \sin d \sin E, \ Z = \cos d,$$ (A8)

where s is a quantity named the "CIO locator", which provides the position of the CIO on the equator of the CIP corresponding to the kinematical definition of the NRO in the GCRS. Its expression as a function of the coordinates X and Y is:

$$s(t) = -\int_{t_0}^{t} \frac{X(t)Y(t) - Y(t)X(t)}{1 + Z(t)} dt - (\sigma_0 N_0 - \Sigma_0 N_0),$$ (A9)

where $\sigma_0$ and $\Sigma_0$ are the positions of the CIO at J2000 and the x-origin of the GCRS, respectively, and $N_0$ is the ascending node of the equator at J2000 in the equator of the GCRS.

The observation zenith angle $\alpha$ and the solar incident angle $\beta$ are as follows:

$$\alpha = \arccos\left(\frac{(X_m - X_T)X_T + (Y_m - Y_T)Y_T + (Z_m - Z_T)Z_T}{\sqrt{((X_m - X_T)^2 + (Y_m - Y_T)^2 + (Z_m - Z_T)^2)(X_T{}^2 + Y_T{}^2 + Z_T{}^2)}}\right),$$ (A10)

$$\beta = \arccos\left(\frac{(X_s - X_T)X_T + (Y_s - Y_T)Y_T + (Z_s - Z_T)Z_T}{\sqrt{((X_s - X_T)^2 + (Y_s - Y_T)^2 + (Z_s - Z_T)^2)(X_T{}^2 + Y_T{}^2 + Z_T{}^2)}}\right),$$ (A11)

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
