# Peer review of "Analysis of Long-Term Moon-Based Observation Characteristics for Arctic and Antarctic"

_remotesensing, doi:10.3390/rs11232805_

Round 1

Reviewer 1 Report

The paper describes how an optical sensor, located in Moon, would be able to observe Earth's polar regions. This should be said clearer in the abstract and in the introduction. Now reader is wondering what are "Moon-based observations". The paper is thorough in describing all the reference systems and details in the observation geometry related to the problem. The biggest problem I had, was with the language of the paper and the minuscule figures. It was also unclear, why the polar sea ice variations were brought up in the discussion. If that is the primary goal, observing sea ice, it should be in introduction, or if it is one possibility it should have it's own chapter, Applications, or similar.

Some minor remarks. 1) You mention Tibet in the beginning of the abstract and never since. I was looking for its re-appearance throughout the whole paper, and it never did. You can justify the importance of polar regions without Tibet, please do so. 2) The Moon, Earth and Sun are written with varying capital letter. Choose one way and stick to it. 3) The figures and their texts are very, very small. It is impossible to see the details of Figure 3 that are described in the text and also all other figures should have text with point size similar to chapters' text. 4) The English is somewhat unclear and sentences were quite long. A revision is needed.

Reviewer 2 Report

The structure of the paper is very good. The argument interesting and original. The content is significant and very well presented.

The discussion and results very clear and appropriate. 

The topic could be of high interest for the readers.

Some notes:

rows:

105        I couldn't find any reference to a 566th Workshop of Xiangshan Scientific Conference in June 2016. Is the reference correct? 

185        correct: "The second type is a reference system dealing with the parameters..."

261    the Authors declare that "for true observations, cloud cover and atmospheric refraction have a huge impact on the observation result, but your analysis didn't take into account both". Can you describe how this can affect the results and at which level this could invalidate them?

281  symmetrical in the north and south Emisphere

pages 14/15 All the diagrams of Figure 9 are not readable at this scale, numbers and writings are fully not readable - please magnify the scale

From row 508 to 528 there is a separation in rows of the last words, probably a problem of alignments.

Reviewer 3 Report

I am very happy to review this well-written, readable, and logically organized paper. The mathematics, logics, and corresponding results are straightforward. I think that this study is crucial to be ready for Moon-based Earth observation. However, a revision process is required before acceptance. The author can see comments as below:

1) The first paragraph of the Introduction

1-1) I know that this paper focused mainly on the Moon-based observation subject for polar regions. It is a new concept to observe various nature in the polar region so that authors need to make a further review of previous Earth-orbital satellite observation of the Earth's environment.

1-2) The authors mentioned two weak points of Earth-orbital satellite observations. However, I think that it is a very small challenge these days. Satellite technology has been developed with accelerating speed, and therefore such kind of problem may not lead to unstable quality of satellite products. For instance, papers published on Remote Sensing special issue of “Remote Sensing of Essential Climate Variables and Their Applications” showed clear large-scale scientific phenomena over Arctic sea ice with well-calibrated passive microwave data. Indeed, Earth-orbital satellites have a problem to observe the Arctic and Antarctic regions that most of satellite is a polar-orbiting satellite, and therefore temporal coverage is weak (Authors can find other papers further). Also, the geostationary satellite cannot make stable data for the Arctic and Antarctic region because of two reasons, one for high zenith angle and the other for cloud presence. Therefore, I recommend that the author change such kind of content following the above comment.

Berg W., R. Kroodsma, C. D. Kummerow, and D. S. McKague, 2018: Fundamental climate data records of microwave brightness temperature, Remote Sensing, 10, 1306.

Lee, S.-M., B.-J. Sohn, and C. D. Kummerow, 2018: Long-term Arctic snow/ice interface temperature from Special Sensor for Microwave Imager measurements, Remote Sensing, 10, 1795.

2) Line 153

Please define JPL. The authors can define it in the first-show Jet Propulsion Laboratory in Line 114.

3) Line 159

Define DE. It should be Development Ephemeris.

4) Caption of Figures

Many scientific journals recommend that the caption should describe the figure itself alone.

5) Figures 5, 6, and 7

There are no titles of x-axes.

6) Figures 6 and 8

I can’t look deeply into scientific features shown in Figure 8 because of its quality. Additionally, the figures make me (or possible reader) confused to understand its scientific meaning, especially for Figure 8. Figure 8 may be summarized by using a table.

7) Figure 9

The figure is very messy. there is no indicator of latitude information in the plot. Please express the order of the pictures.

And lines 474

“VEA can cover the 0 – 33 interval”

I don’t know whether my understanding of figure 9 is correct or not. The VEA never exceeds 30 deg in the top third column (if it corresponds to 85N case) of Figure 9.

9) Lines 511-512

If certain dataset is used in this study, it should be referred to.

Round 2

Reviewer 1 Report

Regarding the terminology "moon-based earth observations" is mostly used by these same authors in their previous work, thus the term is not very widely known or used, but maybe it will be in the future. After the clarifications and text editing the message is much clearer.

Regarding the figures, it is impossible to tell which figures are old and which are new in some cases, e.g. fig. 7 and fig 9. Text is in some cases bigger, and sometimes clearer, but maybe the editor can decide what is of adequate quality.

And I would still call chapter 4. something other than "discussion" as the results are actually discussed in chapter 3 and you are showing a possible application in chapter 4. But I leave that to the editor as well.
